# Apoptotic Cell Death in an Animal Model of Virus-Induced Acute Liver Failure—Observations during *Lagovirus europaeus*/GI.2 Infection

**DOI:** 10.3390/ijms25020798

**Published:** 2024-01-08

**Authors:** Dominika Bębnowska, Rafał Hrynkiewicz, Sylwia Rzeszotek, Marika Freus, Agata Poniewierska-Baran, Paulina Niedźwiedzka-Rystwej

**Affiliations:** 1Institute of Biology, University of Szczecin, Felczaka 3c, 71-412 Szczecin, Poland; rafal.hrynkiewicz@usz.edu.pl (R.H.); agata.poniewierska-baran@usz.edu.pl (A.P.-B.); 2Department of Histology and Embryology, Faculty of Medicine and Dentistry, Pomeranian Medical University in Szczecin, Powstańców Wlkp. 72, 70-111 Szczecin, Poland; sylwia.rzeszotek@pum.edu.pl (S.R.); marika.freus@pum.edu.pl (M.F.)

**Keywords:** *Lagovirus europaeus*/GI.2, Rabbit Hemorrhagic Disease (RHD), apoptosis, rabbits, animal model

## Abstract

*Lagovirus europaeus*/GI.2 causes severe and highly fatal Rabbit Hemorrhagic Disease (RHD). Because of its characteristics, this infection is used as an animal model for acute liver failure (ALF). Apoptosis is one of the key processes underlying ALF and has been described as one of the mechanisms of RHD pathogenesis. Apoptotic cell death has been quite well characterized in infection with different variants of GI.1 strains, but so far, the GI.2 genotype has not been widely studied. In this study, we performed an evaluation of apoptotic cell death in hepatocytes of rabbits infected with *Lagovirus europaeus*/GI.2. We analyzed the expression of genes involved in apoptotic cell death by real-time PCR and performed immunohistochemical (IHC) assays. We showed a significant increase in the expression of *caspase-3* and the proapoptotic *Bax* and anti-apoptotic *Bcl-2* in infected animals. In addition, we recorded increased *Bax*/*Bcl-2* ratios. IHC analyses showed the presence of morphological signs of apoptosis in the hepatocytes of infected rabbits. Our results indicate that caspase-3 and proteins from the Bcl-2 families play a key role in apoptosis induced by *Lagovirus europaeus*/GI.2 infection.

## 1. Introduction

Acute liver failure (ALF) is a condition that is defined by the development of severe acute liver injury accompanied by encephalopathy and impaired synthetic function. The state affects patients who do not have cirrhosis or pre-existing liver disease. The factors leading to the development of ALF can be varied, but the two main causes are drug-induced damage and viral infections, particularly viral hepatitis [1]. The incidence of ALF due to viral infection depends on the geographical region. With the implementation of prophylactic vaccination in Europe, the number of cases of ALF of viral origin has decreased significantly, but in Asia, Africa, and South America, viral hepatitis is still the most common cause of acute liver failure [2]. In general, ALF occurs when the degree of hepatocyte death is higher than the regenerative capacity of the liver. The hepatocyte death observed in ALF can be induced by different pathways of programmed cell death, specifically apoptosis, autophagic cell death, or necrosis [3].

Apoptosis is one of the processes that play a key role in maintaining tissue homeostasis by keeping a balance between proliferation and the removal of damaged, infected, and transformed cells. This process is particularly important for the liver, which is vulnerable to damage through its functions [3]. The specific morphological features of apoptosis include cell deformation, loss of cell membrane integrity, DNA degradation, and the formation of apoptotic bodies. The mechanism of apoptotic cell death is a whole cascade of energy-dependent molecular events. In general, there are two primary apoptotic pathways: an external, which is mediated by the death receptor and an internal, which is a mitochondrial pathway. There is also an additional route that involves cytotoxicity of T lymphocytes and perforin- and granzyme-dependent cell death. As a result of specific triggering signals in the apoptotic cascade, the relevant initiator caspase (-8, -9,-10) is activated, which then activates the executive caspases (-3, -6, -7). The moment of activation of the executive caspases is the focal point from which all apoptotic pathways share a further trail [4].

Apoptotic cell death is one of the antiviral mechanisms involved in host immune defense aimed at limiting the spread of viral particles. In some viral infections, such as hepatitis C virus (HCV), or hepatitis B virus (HBV), this process plays an important role in disease pathogenesis, which is why it represents such an important line of research [5]. One of the viral immunogenic models of ALF is Rabbit Hemorrhagic Disease (RHD), whose etiological agent is *Lagovirus europaeus*/GI.1 and GI.2. This disease is characterized by severe hepatitis with necrosis and disseminated intravascular coagulation (DIC), leading to high mortality [6]. Because of the physiological, histological, and biochemical features observed, RHD fulfils most of the requirements of an animal model of acute liver failure. Because of this, it has been widely used to gain a better insight into the pathogenesis of ALF and to search for agents with hepatoprotective effects [7,8,9]. The application of Rabbit Hemorrhagic Disease as a research model has been extensively described in our earlier work [1].

Within RHD-associated *Lagovirus europaeus*, genotypes (GI.1 and GI.2) and variants (GI.1a-GI.1d) can be distinguished, with GI.1a being the first isolated and best studied of these. In addition to its genetic differences, it is also characterized by a different antigenicity than the other GI.1 strains [10]. The GI.2 genotype has been described relatively recently and is characterized by several important features. First of all, a tendency to cross the species barrier is recorded, as well as the ability to infect rabbits under four weeks of age, whereas the GI.1 strain had an age limit of nine weeks. Moreover, rabbits vaccinated against GI.1 strains are susceptible to GI.2 infection [6]. Studies have shown that RHD induced by different viral strains generates differences in the host immune profile, in particular regarding cellular and humoral immunity, but also apoptotic cell death. For example, it was previously shown that infection with different GI.1 strains results in different percentages of apoptotic granulocytes and lymphocytes, as well as different timelines of recording these changes [11,12,13]. Moreover, we have recently shown that recombinant strains also show a specific pattern of immunological parameters [14]. This indicates that features of *Lagovirus europaeus*/GI.1 such as antigenic variation and hemagglutination capacity influence the immune response, but it is not clear what the immune picture is after infection with strain GI.2. The main organ where apoptotic changes are recorded in RHD is the liver because it is the target site of viral replication [15]. Apoptosis in *Lagovirus europaeus*/GI.2 infection has not been widely analyzed so far [16]. Considering the previously mentioned specific characteristics of GI.2 strains, as well as the different disease course attributed to these strains, such as a longer incubation period and lower mortality rates [6], this study aimed to analyze apoptotic cell death in the livers of rabbits infected with *Lagovirus europaeus*/GI.2.

## 2. Results

### 2.1. Clinical Results and Postmortem Analysis

Typical characteristics of RHD were recorded postmortem in all animals. Observation of clinical signs during the experiment indicated that some animals did not show symptoms of infection (n = 6), and death occurred suddenly at 12–36 h p.i., generating a survival rate of 40% at 36 h p.i. In the remaining animals (n = 4), clinical symptoms indicated outcomes of an acute form of the disease (n = 4). From 24 h p.i., apathy, conjunctival congestion, dyspnoea, body temperature > 41 °C, and anorexia were noted. Humane endpoints were achieved at 36–48 h p.i.

### 2.2. Viral Genome Copies 

To confirm the effectiveness of the infection protocol as determined by the level of viral load, we analyzed the copy number of the viral genome in liver samples (30 mg). We demonstrated the presence of the virus genome in all animals from the experimental group (n = 10). The results ranged from 3.14 × 10^6^ to 2.70 × 10^10^ copies/g of liver (Table 1).

### 2.3. Lagovirus europaeus/GI.2 Infection Promotes Apoptotic Genes (Caspase-3, Bax and Bcl-2) Expression

To assess whether apoptotic gene regulation occurs during *Lagovirus europaeus*/GI.2 infection, we performed an analysis of selected molecular markers (Figure 1, Table 2). Our results indicate that there is a significant upregulation (*p* ≤ 0.03) of caspase-3 in the livers of infected rabbits. This is an executive caspase in apoptosis and is involved in both the mitochondrial and death receptor-mediated pathways. We also assessed the expression of genes encoding the Bcl-2 family of proteins, which are regulators of apoptosis. We recorded significantly increased expression of proapoptotic *Bax* (*p* ≤ 0.01), as well as anti-apoptotic *Bcl-2* (*p* ≤ 0.05) in the infected group as compared with the control group. In general, the *Bax*/*Bcl-2* ratio indicates activation or inhibition of apoptosis. Our analysis showed a significantly increased ratio (*p* ≤ 0.05) of *Bax* to *Bcl-2* expression, indicating a strong activation of apoptosis in *Lagovirus europaeus*/GI.2 infection.

### 2.4. Percentage of the Positive Area with Apoptosis Markers in the Liver Using the VectorImpress Set of Reagents

Figure 2 shows representative photomicrographs comparing expression of apoptosis markers between control and experimental groups using two different antibodies directed against caspase-3 (procaspase-3 and the active form of caspase-3) and one antibody to annexin V within hepatocytes in the livers of rabbits using a VectorImpress set of reagents. The top row shows the negative control, without primary antibodies. The percentage of immunopositive cells was determined by using an algorithm to convert into numerical values, which are presented in Figure 3.

Statistically significant changes in caspase-3 expression were observed between the control and experimental groups in both anti-caspase antibodies. Despite the lack of statistical significance in the case of annexin V in tissues stained with VectorImpress kits, there was a tendency of increased level of expression of annexin V in the livers of animals from the experimental group.

### 2.5. Percentage of the Positive Area with Apoptosis Markers in the Liver Using EnVision Set of Reagents

In order to gain more insight and confirm the reliability of the results, we also carried out the analysis using a different set of reagents. Figure 4 shows the representative photomicrographs comparing expression of apoptosis markers between control and experimental groups using two different antibodies directed against caspase-3 and one antibody to annexin V within hepatocytes in the livers of rabbits using an EnVision set of reagents. The top row shows the negative control, without antibodies. The percentage of immunopositive cells was determined using an algorithm and converted into numerical values, which are presented in Figure 5.

Statistically significant changes in caspase-3 and annexin V expression were observed between the control and experimental groups.

## 3. Discussion

Programmed cell death (PCD) is a process involved in many events in the body to maintain homeostasis. Its initiation occurs in response to genotoxic, metabolic, and infectious agents. For this reason, PCD plays an important role in the pathogenesis of many conditions associated with immune disorders, neurodegeneration, tumorigenesis, and the pathogenesis of infectious diseases. As one type of PCD, apoptosis is an integral part of the host’s defense machinery in response to an invading intracellular pathogen, such as a virus, to limit the production and spread of progeny virions [17]. The triggering of apoptotic cell death can be initiated by an extrinsic or intrinsic pathway at any stage of infection, from the binding of the virus to a receptor on the cell membrane through the penetration of the pathogen into the cell and the subsequent integration of the viral nucleic acid into the host genome [18]. However, some viruses deliberately manipulate apoptosis to promote disease expansion and progression. For example, Human Papillomavirus (HPV) can influence apoptosis through the E6 protein, which can block apoptotic flow via the extrinsic and intrinsic pathways by binding to FAS-associated death domain protein [19], procaspase-8 [20], but also Bax [21]. Inhibition of cell death can also be mediated by viral homologs of Bcl2 (v-Bcl2), which are common in several taxa such as *Poxviridae*, *Herpesviridae*, *Asfarviridae,* and *Iridoviridae*. The direct action of v-Bcl2 is to inhibit the oligomerization of Bax and Bad, which blocks the release of cytochrome c from mitochondria, or attenuates signals by inactivating proapoptotic proteins [17]. Moreover, Ren et al. [22] recently reported that the exon-1 UL37 (UL37x1) protein encoded by human cytomegalovirus (HCMV), in addition to inhibiting apoptosis, also inhibited the cGAS-STING pathway, which is important in viral infections because its activation leads to the expression of genes encoding proteins such as interferons, which consequently inhibit host innate immunity. On the other hand, the interaction of the virus with the host cells may have the effect of enhancing apoptosis. Medigeshi et al. [23] reported that West Nile Virus (WNV) imposes endoplasmic reticulum (ER) stress through increased production of viral proteins, leading to the induction of the unfolded protein response (UPR) pathway and subsequent apoptosis. Furthermore, Wang et al. [24] showed that non-structural protein 4B (NS4B) of Japanese encephalitis virus induced dimerization and activation of the ER stress mediator PERK (RNA-dependent protein ER kinase), resulting in neuronal apoptosis. For some viruses, the question of the effect of the virus on apoptosis is not fully clear. Studies have shown that HCV viral proteins can exhibit both proapoptotic and anti-apoptotic effects. Javed and Manzoor [25] reported that the non-structural NS4A protein of HCV genotype 3a induced cell death through the mitochondrial pathway triggered by Bax while down-regulating the anti-apoptotic protein Bcl-xL. In contrast, in a study by Erdtmann et al. [26], the NS2 HCV protein interfered with the death pathway induced by the liver-specific CIDE-B protein, a member of the Cell Death-Inducing DNA Fragmentation Factor α (DFFA)-like Effector (CIDE) family of proapoptotic proteins, which mediates cytochrome c release from mitochondria and caspase-3 activity.

Programmed cell death has been described as an important part of the pathogenesis of RHD in *Lagovirus europaeus*/GI.1 infection. Studies have shown that apoptosis affects not only the liver, but also the kidney, spleen, lung, intravascular macrophages, and endothelial cells [9,15]. Furthermore, Alonso et al. [15] described that, given that apoptotic cells are locations where increased procoagulant activity occurs, apoptosis of this population of macrophages and endothelial cells may represent the first step in the pathogenesis of the DIC syndrome observed in Rabbit Hemorrhagic Disease. Apoptosis has also been extensively analyzed in peripheral blood lymphocytes and granulocytes in infection with different GI.1 strains, and it has been shown that there are huge differences in the intensity and dynamics of this process, as well as in the group of most affected cells, depending on the biological characteristics of the strain [11,13]. Moreover, we recently reported that peripheral blood lymphocyte apoptosis induced by cytotoxic T lymphocytes (CTL) occurs in *Lagovirus europaeus*/GI.1a infection, and this is a new element in the pathogenesis of RHD [12]. 

Our results indicate that in *Lagovirus europaeus*/GI.2 infection there is a strong activation of apoptotic cell death in the liver. To investigate apoptosis, we also performed immunohistochemical analysis using two independent reagent kits. The results confirmed that significant expression of caspase-3 occurs in infection. We also assayed annexin V to detect morphological hallmarks of apoptosis in hepatocytes. Hence, our study demonstrates strong apoptosis in the livers of rabbits infected with *Lagovirus europaeus*/GI.2, highlighting the role of this process in the pathogenesis of RHD caused by strain GI.2. Our results remain consistent with existing reports, as Garcia-Lastra et al. [27] showed that in *Lagovirus europaeus*/GI.1 infection, caspase-3 activity significantly increased in the liver at 36 and 48 h after infection of the animals. Similarly, in a study by Chen et al. [28], the viral non-structural protein NSP6 induced caspase-3 activity in the rabbit kidney cell line RK13. Moreover, it was reported that NSP6 induced apoptosis through death receptors, as indicated by increased activity of caspase-8 and the mitochondrial pathway as confirmed by increased caspase-9 expression. However, in contrast, a study by Vallejo et al. [29] showed that apoptosis in RHD induced by strain GI.1 occurred without a significant increase in caspase-3 activity.

In our study, we also monitored molecular markers of apoptosis via the mitochondrial pathway. We showed that the proapoptotic *Bax* and the anti-apoptotic *Bcl-2* are significantly up-regulated in infection. Analysis of the *Bax/Bcl-2* ratio showed a significant up-regulation of the proapoptotic protein. Similar to the results obtained in our study, it was also recorded that *Lagovirus europaeus*/GI.1 infection was accompanied by apoptosis associated with up-regulation of *Bax* and *Bcl-2* genes, as well as elevated *Bax/Bcl-2* ratios [28,30]. However, Vellejo et al. [29] showed that in infection with strain GI.1, there was inhibition of anti-apoptotic *Bcl-2* and *Bcl-xL* expression at 30 and 36 h post-infection. These data indicate that the molecular pathways of cell death should be extensively analyzed in *Lagovirus europaeus* infections, as there may be key differences in the level of expression regulation of individual signaling pathway elements dependent on the virus strain.

Targeting elements involved in apoptotic cell death are an important part of ALF treatment research, so understanding the molecular basis of the signaling caspase is a crucial step. The results obtained in this study indicate that genes encoding caspase-3, Bax and Bcl-2 are strongly involved in apoptotic death in *Lagovirus europaeus*/GI.2 infection and could be considered as potential therapeutic targets and perhaps also prognostic factors, so further studies verifying this hypothesis seem warranted. Furthermore, caspase-3 seems to be a key element involved in apoptosis induced by *Lagovirus europaeus*/GI.2 infection, so it is worthwhile to consider in the future the effect of inhibitors of this caspase on the course and progression of pathology in rabbits with RHD. 

## 4. Materials and Methods

### 4.1. Ethical Statement

The experimental infection of animals was carried out at the Pomeranian Medical University in Szczecin based on the consent obtained by the Local Ethical Committee for Experiments on Animals in Poznań No. 35/2022. Rabbits were maintained in accordance with the European Union and national guidelines for animal experimentation. The experiment was conducted in accordance with the ARRIVE guidelines. 

### 4.2. Scheme of the Experiment

European rabbits Oryctolagus cuniculus/Crl:KBL (NZW)/0052 were used in the study. The animals were not vaccinated against *Lagovirus europaeus*/GI.1 and GI.2 and were purchased from a licensed breeder. The experiment included twenty rabbits (6 months old) with body weights in the range of 4.5 kg (±10%). The animals were randomly divided into a control group (n = 10) and an experimental group infected with the *Lagovirus europaeus*/GI.2 strain (n = 10). In the control and experimental groups, 50% were males and 50% females. After delivery to the laboratory, the animals were adapted over a period of three weeks. Hay balls and wooden blocks were placed in the cages as environmental enrichment. The animals were kept under standard laboratory conditions with a 12 h light/12 h dark cycle, a temperature of 19 ± 2 °C, a humidity level of 50 ± 10% and 15–20 air changes per hour. 

The virus strain used in the experiment was derived from a rabbit that naturally died in Italy (RHDV2_Ri2017 strain; GenBank: OQ680671). Rabbits in the experimental group were infected by intramuscular injection of an inoculum prepared according to the method described earlier [12]. The inoculate (1 mL) contained 2 × 10^4^ hemagglutination units of the virus, which was determined by the hemagglutination assay. Animals in the control group received intramuscular injections of the same amount (1 mL) of placebo containing PBS (phosphate-buffered saline).

The start of the experiment (marked as 0 h of the experiment) occurred at the time of inoculum administration. After inoculation, the health status of all animals, including body temperature measured rectally, was monitored at least twice daily. The onset of severe symptoms of rabbit hemorrhagic disease was the final moment of the experiment, after which each qualified animal was euthanized. Animals qualified for euthanasia were anesthetized by intravenous administration of a preparation of sodium pentobarbital at 90 mg/kg, followed by administration of the cardiac arrest-inducing preparation of sodium pentobarbital at 250 mg/kg. Tissue samples for the study were collected from rabbits immediately post-mortem and were preserved in RNAlater RNA Stabilization Solution (Thermo Fisher Scientific, Waltham, MA, USA) and stored at −80 °C until analysis.

### 4.3. Quantitative Real-Time PCR (qRT-PCR)

Total RNA was isolated from frozen liver samples (30 mg) using the RNeasy Mini Kit (A&A Biotechnology, Gdańsk, Poland) according to the manufacturer’s recommendations. Elution was performed in 100 μL. The quantity and purity of the isolates were analyzed using the NanoDrop™ 2000 spectrophotometer (Thermo Fisher Scientific, Waltham, MA, USA). Reverse transcription of RNA isolates (in the amount of 200 ng determined on the basis of the concentrations) was performed using Transcriptor First Strand cDNA Synthesis Kit (Roche Diagnostics GmbH, Mannheim, Germany) as recommended by the manufacturer. Quantitative real-time PCR was performed using the LightCycler^®^ 480 Instrument II (Roche Diagnostics GmbH, Mannheim, Germany) and the PowerUp™ SYBR™ Green Master Mix (Thermo Fisher Scientific, Waltham, MA, USA) according to the manufacturer’s instructions. In the study, specific primers were used [28,31]: *Caspase-3*—F(5′-GCTGGACAGTGGCATCGAGA-3′) and R (5′-TCCGAATTTCGCCAGGAATAGTAA-3′), *Bax*—F (5′-GTGTCTCAAGCGCATTGGCG-3′) and R(5′-CAAACATGTCGGCCTGCCACT-3′), *Bcl-2*—F(5′-GATTGTGGCCTTCTTTGAGTTC-3′) and R (5′-GGCAGCGATCATCCTCTGTA-3′), *β-actin*—F(5′-TGGCATCCTGACGCTCAA-3′) and R (5′- TCGTCCCAGTTGGTCACGAT-3′). The threshold cycle (Ct) values were normalized by *β-actin* gene expression, and the relative gene expression was calculated using the Pfaffl method [32].

Additionally, a dilution series of full-length *L. europaeus* GI.2 transcript standards was prepared ranging from 1 × 10^6^ copies/μL to 1 × 10^2^ copies/μL. A curve was prepared using a LightCycler 480 II instrument and used for absolute quantification to determine the copy number of *L. europaeus* GI.2. In the study, specific primers were used: RHDV2 F(5′-TGGAACTTGGCTTGAGTGTTGA-3′) and R(3′-ACAAGCGTGCTTGTGGACGG-5′) [33].

### 4.4. Tissue Preparation for Immunohistochemical Reactions

The dissected livers were fixed in 4% formalin for at least 24h, but no more than 36 h, and subsequently washed with absolute ethanol (3 times over 3h), absolute ethanol with xylene (1:1) (twice over 1h), and xylene (3 times over 20 min). Then, after 3 h of tissue saturation with liquid paraffin, the samples were embedded in paraffin blocks. Using a microtome (Microm HM340E, Thermo Fisher Scientific, Walldorf, Germany), 3–5 μm serial sections were taken and placed on poly-L-Lysine microscope slides (Thermo Scientific, Leicestershire, UK; cat. no. J2800AMNZ). The sections of the livers were deparaffinized in xylene and rehydrated in decreasing concentrations of ethanol, and then used for IHC reactions.

### 4.5. Immunohistochemical Methods

For IHC reactions we used two sets of reagents: EnVision FLEX Mini Kit, High pH (Agilent, Santa Clara, CA, USA; cat. no. K8023) and ImmPRESS HRP Universal (Horse Anti-Mouse/Rabbit IgG) PLUS Polymer Kit, Peroxidase (Vector Laboratories, Newark, CA, USA; cat. no. 7800). To expose the epitopes to the IHC procedure, the deparaffinized and rehydrated sections were boiled in Target Retrieval Solution (DakoCytomation, Glostrup, Denmark, cat. no. S2369) in a water bath (99 °C for 8 min.). Once cooled and washed with PBS, the endogenous peroxidase activity was blocked using peroxidase blocking reagent included in the sets of reagents, and then the slides were incubated for 90 min. at room temperature with primary antibodies against procaspase-3 (Abcam, Cambridge, UK, cat. no. ab184787, final dilution 1:200), annexin V (Santa Cruz Biotechnology, Dallas, TX, USA, cat. no. sc-393669, final dilution 1:200), and the active form of caspase-3 (Novus Biologicals, Centennial, CO, USA, cat. no. NB600-1235, final dilution 1:200). Antibodies were diluted in Antibody Diluent with Background Reducing Components (Dako, Dako, Santa Clara, CA, USA, S302 S3022).

To visualize the antigen-antibody complex, two sets of reagent HRP-related systems were used, based on the reaction of avidin-biotin-horseradish peroxidase with DAB as a chromogen, according to the included staining procedure instructions. Sections were washed in distilled H_2_O and counterstained with hematoxylin (Mar-Four, Konstantynów Łódzki, Poland). For a negative control, specimens were processed in the absence of primary antibodies. Positive staining was determined microscopically (Leica DM5000B, Wetzlar, Germany) by visual identification of brown pigmentation. The experiments were repeated independently twice.

### 4.6. Quantitative Computer Image Analysis Histological Slides

Stained with antibodies against caspase-3 and annexin V, slides were scanned at a magnification of 40× (resolution of 0.25 μm/pixel) using the GRUNDIUM OCUS^®^40 (Grundium, Tampere, Finland). The obtained digital images of the slides were analyzed using the slide viewer 3DHISTECH (version 2.5; Budapest, Hungary) with support of the image analysis platform QuantCenter (PatternQuant, version 2.3, Budapest, Hungary).

For the automatic computer analysis of positive signals (brown pigmentation) in rabbit livers, examples of positive and negative cells were marked and then the algorithm counted the percentage of positive area of the tissues. Other parameters were set to achieve compliance with the visual evaluation of color intensity, including the intensity thresholds for positive results. The total number of immuno-positive areas was counted in random fields in each studied group with an average area of 4 mm^2^ (5 fields per rabbit). IHC staining and analysis were conducted in two independent experiments and data were summarized. The experiments were repeated independently twice.

### 4.7. Statistical Analysis

The statistical analysis was performed using Tibco Statistica 13.3 (StatSoft, Palo Alto, CA, USA). The values of the parameters were presented as arithmetic means, standard deviations (SD), and standard error (SE). The normality of variable distributions was verified by the Shapiro–Wilk Test. Data with a normal distribution were analyzed using the Student’s t-test and the Mann–Whitney U test was used for data with a non-normal distribution.

## 5. Conclusions

In summary, our study demonstrates the activation of apoptotic cell death in the liver of rabbits infected with *Lagovirus europaeus*/GI.2, and this process underlies the damage to hepatocytes in RHD induced by strain GI.2. Undoubtedly, terminal caspase-3 plays a key role in the apoptotic response discussed above. This study indicates that the mitochondrial apoptotic pathway is activated during infection with upregulation of proapoptotic Bax and anti-apoptotic Bcl2. However, further studies will be needed to fully understand the issue of programmed cell death in the livers of rabbits infected with *Lagovirus europaeus*/GI.2, as we have not included markers of the death receptor-mediated pathway in this work, and this is a limitation. Given that some viral-induced ALF, such as HCV or HBV infection, involves the killing of infected hepatocytes by cytotoxic T lymphocytes (CTLs) through Fas receptor-mediated apoptosis [34], it may be of interest to determine the role of CLT cells in hepatocyte apoptosis induced by *Lagovirus europaeus*/GI.2 infection. Another caveat is that our study only includes liver tissue but does not investigate other tissues and organs, which should also be a direction for future research because RHD affects multiple systems in the host body.

## Figures and Tables

**Figure 1 ijms-25-00798-f001:**
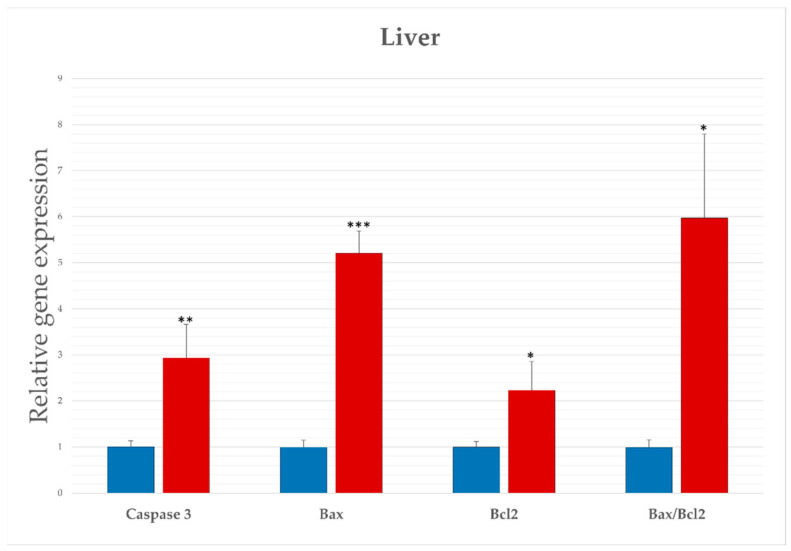
Expression levels of apoptotic parameters in rabbits infected with *Lagovirus europaeus*/GI.2. Real-time PCR results showed that expression levels of all the genes were higher in the experimental group than in the control group. Data are presented as mean and SE. Statistical significance is marked as * *p* ≤ 0.05 ** *p* ≤ 0.03; *** *p* ≤ 0.01.

**Figure 2 ijms-25-00798-f002:**
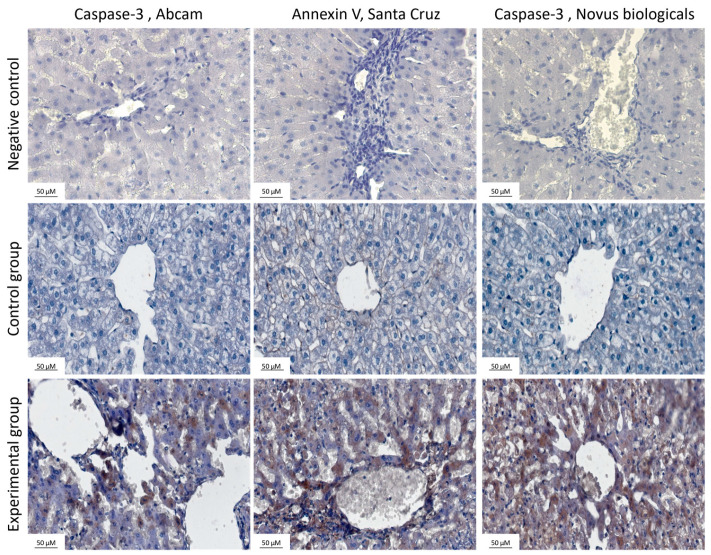
Livers stained with markers for apoptosis: procaspase-3 (Abcam, Cambridge, UK), **left** column, annexin V (Santa Cruz, Biotechnology, Dallas, TX, USA), **middle** column, active form of caspase-3 (Novus Biologicals, Centennial, CO, USA), **right** column. The top row shows the negative control. The middle row shows sections from the control group and the bottom row shows sections from the experimental group. Magnification applied ×40. Pictures taken by Leica (DM5000B, Wetzlar, Germany). Slides were stained with a VectorImpress set of reagents. The experiments were repeated independently twice (n = 10).

**Figure 3 ijms-25-00798-f003:**
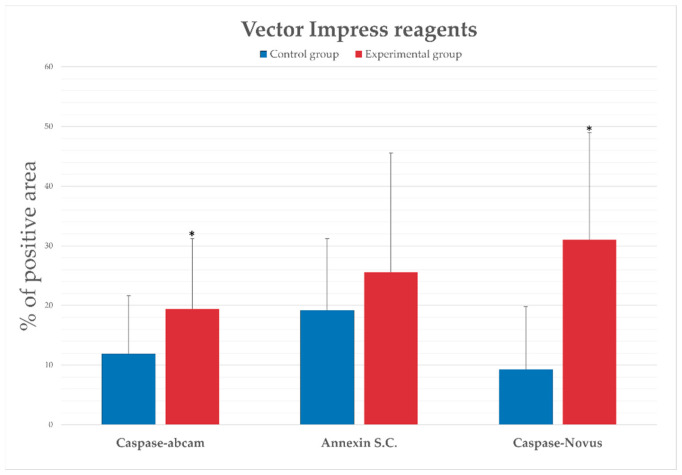
Percentage of the immunopositive area obtained from computer analysis using 3DHISTECH (version 2.5; Budapest, Hungary) with support of the image analysis platform for immunohistochemically stained images SlideViewer. A statistically significant difference between experimental and control groups in the expression of procaspase-3 (antibody from Abcam) and between experimental and control groups in the expression of the active form of caspase-3 (antibody from NovusBiologicals) is marked. Data are presented as mean and SD. The experiments were repeated independently twice, and the results obtained were presented in the form of one graph. Slides were stained with a VectorImpress set of reagents. Statistical significance is marked as * *p* ≤ 0.05.

**Figure 4 ijms-25-00798-f004:**
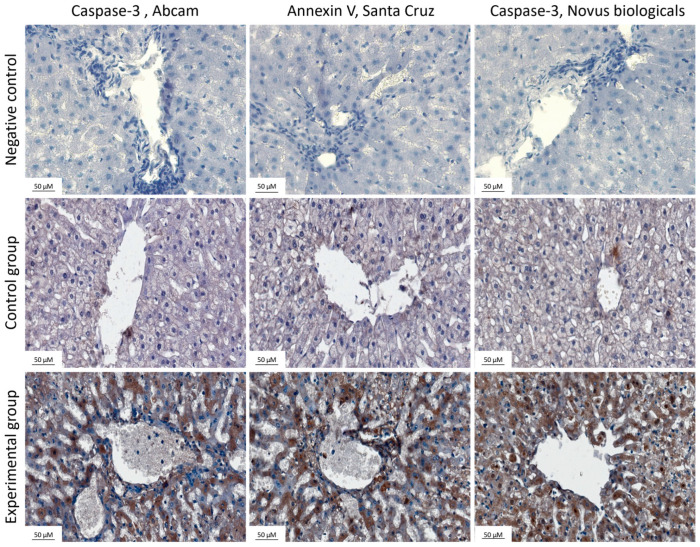
Livers stained with markers for apoptosis: caspase-3 (Abcam), **left** column, annexin V (Santa Cruz), **middle** column, caspase-3 (Novus Biologicals), **right** column. The top row shows the negative control. The middle row shows samples from the control group and the bottom row samples from the experimental group. Magnification applied ×40. Pictures taken by Leica (DM5000B, Wetzlar, Germany). Slides were stained with EnVision set of reagents. The experiments were repeated independently twice (n = 10).

**Figure 5 ijms-25-00798-f005:**
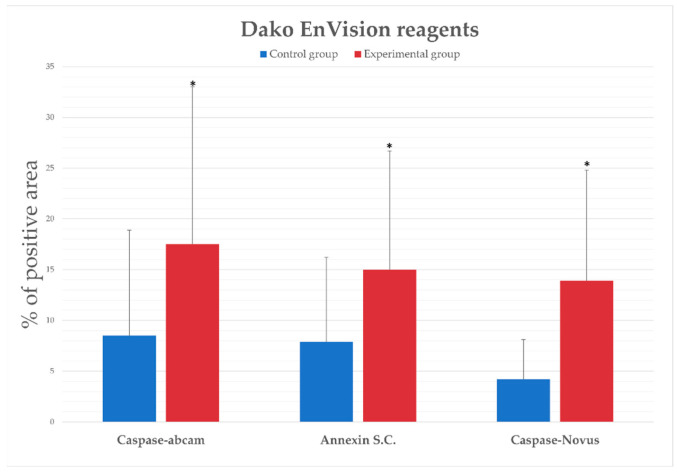
Percentage of the immunopositive area obtained from computer analysis using 3DHISTECH (version 2.5; Budapest, Hungary) with the support of an image analysis platform for immunohistochemically stained images SlideViewer. The differences between experimental and control groups in the expression of caspase-3 (antibodies from Abcam and NovusBiologicals) and between experimental and control groups in the expression of annexin V are marked. Data are presented as mean and SD. The experiments were repeated independently twice, and the results obtained were presented in the form of one graph. Slides were stained with an EnVision set of reagents. Statistical significance is marked as * *p* ≤ 0.05.

**Table 1 ijms-25-00798-t001:** Viral load in liver samples of infected rabbits.

Rabbit	CT	Number of Copies [copies/g]
K1/2	26.55	3.14 × 10^6^
K2/2	23.26	4.45 × 10^8^
K3/2	26.37	6.20 × 10^6^
K4/2	22.20	2.06 × 10^9^
K5/2	23.11	5.46 × 10^8^
K6/2	22.30	1.67 × 10^9^
K7/2	21.79	3.39 × 10^9^
K8/2	20.30	2.70 × 10^10^
K9/2	22.37	1.57 × 10^9^
K10/2	22.60	1.42 × 10^9^

CT—threshold cycle.

**Table 2 ijms-25-00798-t002:** Relative expression of apoptotic markers obtained by qRT-PCR.

Gen	Parameter	Results	
	Group	CG	EG
*Caspase-3*	Mean	1.00	2.94
	SD	0.44	2.29
	SE	0.14	0.72
	*p*-value	0.014 **	
*Bax*	Mean	0.99	5.20
	SD	0.48	1.56
	SE	0.15	0.49
	*p*-value	<0.0001 ***	
*Bcl-2*	Mean	1.00	2.24
	SD	0.39	1.95
	SE	0.12	0.62
	*p*-value	0.0305 *	
*Bax/Bcl2* ratio	Mean	0.99	5.97
	SD	0.53	5.74
	SE	0.17	1.82
	*p*-value	0.0304 *	

CG-control group; EG-experimental group; SD- standard deviation; SE- standard error; * *p* ≤ 0.05 ** *p* ≤ 0.03; *** *p* ≤ 0.01.

## Data Availability

The data presented in this study are available on request from the corresponding author.

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
