# Peer review of "Apoptotic Cell Death in an Animal Model of Virus-Induced Acute Liver Failure—Observations during Lagovirus europaeus/GI.2 Infection"

_ijms, 2024, doi:10.3390/ijms25020798_

Round 1

Reviewer 1 Report

Comments and Suggestions for Authors

The paper is well-written and informative, with results that are clearly presented in the figures and thoroughly described in the manuscript. However, I have two remarks:

  1. The authors should consider adding a description of the use of a rabbit viral infection model for human liver failure. This would enhance the relevance and applicability of the study.

  2. In the methodology, the authors isolated total RNA from 30 mg of liver tissue, followed by quantitative PCR (qPCR) analyses, expressing the viral load in units of copies per gram of liver tissue. However, several clarifications are needed:

    • How were these calculations performed?
    • Did all isolations yield consistent RNA concentrations?
    • On what basis do the authors assume a specific RNA level (mass) in milligrams or grams of liver tissue?

    While using the described units is acceptable, a more extensive description of the methods and calculations would be beneficial. It would be most appropriate to assess isolated RNA concentrations using a spectrophotometer (which was done) and to perform reverse transcription using a defined amount of total RNA (e.g., 500 ng, 1000 ng).

Regarding the sentence on line 291-292, it is somewhat misleading. The NanoDrop device measures RNA concentration and the presence of contaminants, not the quality of the RNA. RNA quality is typically assessed using non-denaturing electrophoresis or a Bioanalyzer (capillary electrophoresis). Therefore, this sentence should be revised for accuracy.

Author Response

Dear Reviewer 1,

On behalf of the authors of the article " Apoptotic cell death in an animal model of virus-induced acute liver failure – observations during Lagovirus europaeus/GI.2 infection”, we would like to thank you for the informative and detailed reviews of our article. We followed all the suggestions and tried to fulfill all their suggestions, and all changes are marked (blue color) in the updated version of the manuscript. Here are the point-by-point answers.

The authors should consider adding a description of the use of a rabbit viral infection model for human liver failure. This would enhance the relevance and applicability of the study.

RE: A summary of the use of RHD as a research model for ALF in humans was discussed in our earlier review article. We have therefore decided to add this information in the Introduction section along with a reference to the aforementioned publication (Bębnowska D, Niedźwiedzka-Rystwej P. The Interplay between Autophagy and Virus Pathogenesis-The Significance of Autophagy in Viral Hepatitis and Viral Hemorrhagic Fevers. Cells. 2022 Mar 3;11(5):871. doi: 10.3390/cells11050871.).

In the methodology, the authors isolated total RNA from 30 mg of liver tissue, followed by quantitative PCR (qPCR) analyses, expressing the viral load in units of copies per gram of liver tissue. However, several clarifications are needed:

How were these calculations performed?

Did all isolations yield consistent RNA concentrations?

On what basis do the authors assume a specific RNA level (mass) in milligrams or grams of liver tissue?

While using the described units is acceptable, a more extensive description of the methods and calculations would be beneficial. It would be most appropriate to assess isolated RNA concentrations using a spectrophotometer (which was done) and to perform reverse transcription using a defined amount of total RNA (e.g., 500 ng, 1000 ng).

RE: We have performed reverse transcription using 200 ng of RNA and this information has been added to the manuscript in the methodology section. We would not like to include information about the calculation method in our manuscript because we do not think this information is important in the methods section, but we are of course open to sharing this information:

Data:

amount of tissue used for isolation (mg)

RNA concentration (measured with NanoDrop) (ng/μl)

elution volume (μl)

amount of RNA used for cDNA synthesis (ng)

cDNA synthesis volume (μl)

amount of cDNA used for real-time PCR (μl)

real-time PCR result (copies/μl)

Example:

amount of tissue used for isolation – 30 mg

RNA concentration – 200 ng/μl

elution volume – 30 μl

amount of RNA used for cDNA synthesis – 100 ng

cDNA synthesis volume – 20 μl

amount of cDNA used for real-time PCR – 1 μl

real-time PCR results = 1000 copies/μl

 =  = 5 ng/μl

(This resulted in 1000 copies/ 5 ng cDNA or 200 copies/ng cDNA)

We got:

RNA concentration – 200 ng/μl and elution volume – 30 μl obtained from 30 mg of tissue

 = 200* 30 = 6000 ng total RNA

6000 ng* 200 copies/ng = 1200000 copies (in 30 mg of tissue)

1200000 copies / 30 mg of tissue = 40000 copies/mg of tissue

We hope that this way of explaining our calculations is clear and understandable.

Regarding the sentence on line 291-292, it is somewhat misleading. The NanoDrop device measures RNA concentration and the presence of contaminants, not the quality of the RNA. RNA quality is typically assessed using non-denaturing electrophoresis or a Bioanalyzer (capillary electrophoresis). Therefore, this sentence should be revised for accuracy.

RE: The indicated sentence has been improved.

Again, we would like to thank you for your effort and time, and we are hoping that our manuscript in its current form will fulfill the requirements of the journal.

Reviewer 2 Report

Comments and Suggestions for Authors

Major Points:

  1. 1. Background and Context:

    • The manuscript could benefit from a more comprehensive background on the different GI strains. Providing insights into the distinctive features of these strains and why apoptosis might vary among them would enhance the contextual understanding for readers.
  2. 2. Discussion of Limitations:

    • While the study sheds light on apoptosis in the liver, it falls short in considering potential apoptotic pathways in other tissues. Addressing this limitation and discussing the broader apoptotic processes across various tissues would strengthen the manuscript.
  3. 3. Discussion of Implications:

    • The implications of the findings could be expanded, especially in terms of the development of treatments targeting apoptosis. Elaborating on the potential applications and relevance of the results to therapeutic interventions would contribute significantly to the discussion.
  4. 4. Clinical Implications:

    • Clarification on the clinical symptoms and outcomes observed in the infected rabbits is warranted. Including this information will provide a more holistic understanding of the disease progression.
  5. 5. Methodological Details:

    • To improve the reproducibility of the study, more details on the methods are needed. This includes information on the virus dose, timing of tissue collection, and the statistical methods employed for data analysis.

Minor Points:

  1. Figure Clarity:
    • Figures would benefit from more descriptive legends to aid readers in interpreting the data. Additionally, defining abbreviations in the figures would enhance clarity.

General Comments:

The manuscript presents valuable insights into apoptosis in GI.2-induced RHD. Addressing the aforementioned points will undoubtedly elevate the quality and impact of the research. The authors are encouraged to carefully consider and incorporate these suggestions to strengthen the manuscript.

Comments on the Quality of English Language

carefully check the manuscript to avoid any spelling or grammer errors

Author Response

Dear Reviewer 2,

On behalf of the authors of the article " Apoptotic cell death in an animal model of virus-induced acute liver failure – observations during Lagovirus europaeus/GI.2 infection”, we would like to thank you for the informative and detailed reviews of our article. We followed all the suggestions and tried to fulfill all their suggestions, and all changes are marked in the updated version of the manuscript. Here are the point-by-point answers. 

Major Points:

  1. Background and Context:

The manuscript could benefit from a more comprehensive background on the different GI strains. Providing insights into the distinctive features of these strains and why apoptosis might vary among them would enhance the contextual understanding for readers.

RE: The background of differences in Lagovirus europaeus strain has been improved. (line: 69-91)

  1. Discussion of Limitations:

While the study sheds light on apoptosis in the liver, it falls short in considering potential apoptotic pathways in other tissues. Addressing this limitation and discussing the broader apoptotic processes across various tissues would strengthen the manuscript.

RE: This limitation of our study has been added in the Conclusions section. (line: 408-410)

  1. Discussion of Implications:

The implications of the findings could be expanded, especially in terms of the development of treatments targeting apoptosis. Elaborating on the potential applications and relevance of the results to therapeutic interventions would contribute significantly to the discussion.

RE: We have added in the discussion section a new paragraph about potential implications of our findings. (line: 270-278)

  1. Clinical Implications:

Clarification on the clinical symptoms and outcomes observed in the infected rabbits is warranted. Including this information will provide a more holistic understanding of the disease progression.

RE: The description of clinical symptoms has been improved. (line: 94-100)

  1. Methodological Details:

To improve the reproducibility of the study, more details on the methods are needed. This includes information on the virus dose, timing of tissue collection, and the statistical methods employed for data analysis.

RE: All the missing information has been added in the methodology section. (line: 301-302; 312-315 and 388-394)

Minor Points:

Figure Clarity:

Figures would benefit from more descriptive legends to aid readers in interpreting the data. Additionally, defining abbreviations in the figures would enhance clarity.

RE: The descriptions of the figures have been improved. We try to be as transparent as possible. We hope that in this form each figure will be better understood.

Again, we would like to thank you for your effort and time, and we are hoping that our manuscript in its current form will fulfill the requirements of the journal.

Round 2

Reviewer 1 Report

Comments and Suggestions for Authors

The paper has been significantly improved and the remarks have been addressed. The paper is ready for sharing on line.

Reviewer 2 Report

Comments and Suggestions for Authors

The authors have addressed my concerns